# Compressed Imaging Reconstruction Based on Block Compressed Sensing with Conjugate Gradient Smoothed *l*_0_ Norm

**DOI:** 10.3390/s23104870

**Published:** 2023-05-18

**Authors:** Yongtian Zhang, Xiaomei Chen, Chao Zeng, Kun Gao, Shuzhong Li

**Affiliations:** 1School of Optics and Photonics, Beijing Institute of Technology, Beijing 100081, China; 2MOE Key Laboratory of Optoelectronic Imaging Technology and System, Beijing Institute of Technology, Beijing 100081, China; 3Luoyang Electro Optic Equipment Research Institute of AVIC, Luoyang 471000, China

**Keywords:** compressed imaging reconstruction technology, block compressed sensing, smooth l0 norm, conjugate gradient method

## Abstract

Compressed imaging reconstruction technology can reconstruct high-resolution images with a small number of observations by applying the theory of block compressed sensing to traditional optical imaging systems, and the reconstruction algorithm mainly determines its reconstruction accuracy. In this work, we design a reconstruction algorithm based on block compressed sensing with a conjugate gradient smoothed $l_{0}$ norm termed BCS-CGSL0. The algorithm is divided into two parts. The first part, CGSL0, optimizes the SL0 algorithm by constructing a new inverse triangular fraction function to approximate the $l_{0}$ norm and uses the modified conjugate gradient method to solve the optimization problem. The second part combines the BCS-SPL method under the framework of block compressed sensing to remove the block effect. Research shows that the algorithm can reduce the block effect while improving the accuracy and efficiency of reconstruction. Simulation results also verify that the BCS-CGSL0 algorithm has significant advantages in reconstruction accuracy and efficiency.

## 1. Introduction

With the development of information technology and the continuous improvement of related requirements in the military and civilian fields, higher standards for the imaging resolution of optical systems have been put forward. The traditional optical imaging system improves resolution mainly by increasing the focal length and aperture of the optical system, reducing the size of detector pixels, and increasing the number of pixels. In practical engineering, the aperture and focal length of optical systems are challenging to improve and limited by observation conditions with stricter requirements. At present, the size of the detector pixels is close to the physical limit, making subsequent improvements difficult. Moreover, increasing the number of detector pixels will increase power consumption, volume, weight, and data storage and processing complexity for the system [1,2]. Therefore, seeking a new image data acquisition and processing method is necessary to improve the imaging system’s resolution.

The emergence of compressed sensing (CS) [3,4] theory provides a new idea for improving the resolution of optical imaging systems. According to CS theory, signals with sparse characteristics can be compressed and projected into a low-dimensional space by a specific observation matrix to obtain a small number of projected signals that can reconstruct the original signal using the corresponding reconstruction algorithm. Compressed imaging super-resolution reconstruction technology applies CS theory to traditional optical systems to achieve super-resolution reconstruction and improve the resolution of imaging systems. This reconstructs higher-resolution images with a small number of observation results. This method has excellent application prospects, reducing the information for image storage, transmission, and processing in optical systems and the requirements of the detector. It transfers the difficulty of improving the system resolution from the hardware to the software side, thus decreasing the equipment cost.

The observation matrix and the reconstruction algorithm determine the reconstruction accuracy of the compressed imaging super-resolution reconstruction technique. The reconstruction algorithm is the primary determinant factor. At present, reconstruction algorithms can be broadly divided into three directions, including the minimization of the $l_{0}$, $l_{1}$, and $l_{p}$ norms. The solving model of the reconstruction algorithm, based on the $l_{0}$ norm minimization, is $\min{∥s∥}_{0},s.t.y=As$, which is an intractable NP−hard problem [5] and must be transformed. The standard methods include the greedy [6] and SL0 algorithms [7]. The reconstruction algorithm, based on the $l_{1}$ norm minimization, uses the $l_{1}$ norm to approximate the $l_{0}$ norm, whose solution model is $\min{∥s∥}_{1},s.t.y=As$. The common methods include the Basis Pursuit algorithm [8] and the Split-Bregman algorithm [9]. The reconstruction algorithm based on the $l_{p}$ norm minimization uses the $l_{p}$ norm to approximate the $l_{0}$ norm. The solution model is $\min{∥s∥}_{p},s.t.y=As$. The commonly used algorithms include the iterative reweighted least squares [10] and the FOCUSS algorithms [11].

Among the three types of algorithms mentioned above, the algorithm based on the minimum $l_{1}$ norm that introduced the convex programming methods has excellent reconstruction precision. However, its time complexity is much larger than others, with refactoring needed over a long time. At the same time, this method requires more observations because the $l_{1}$ norm cannot accurately represent the signal’s sparsity. The algorithm based on the minimum $l_{p}$ norm reduces the number of required observations, but the reconstruction time is still long, and the practicability is not higher. The algorithm based on the minimum $l_{0}$ norm is divided into the greedy and SL0 algorithms. The greedy algorithm requires a small number of calculations and has a fast reconstruction speed, but the reconstruction accuracy is poor, and there are reconstruction errors.

The SL0 algorithm, as a reconstruction algorithm based on the $l_{0}$ norm minimization, approximates the $l_{0}$ norm through the constructor, combines the idea of convex programming with the greedy algorithm, and greatly reduces the reconstruction time while ensuring a high reconstruction accuracy. Therefore, this paper focuses on the improvement of the SL0 algorithm.

Based on the SL0 algorithm, this paper adopts the inverse trigonometric fraction function approximation to express the $l_{0}$ norm and combines the modified conjugate gradient method to improve the SL0 algorithm, which is called the CGSL0 (Conjugate Gradient Smooth l0 Norm) algorithm. On this basis, to eliminate the image block effect after reconstruction, this paper proposes a BCS reconstruction algorithm based on the CGSL0 and BCS-SPL [12] algorithms termed the BCS-CGSL0 (Block Compressed Sensing with Conjugate Gradient Smooth $l_{0}$ Norm) algorithm, combining the CGSL0 and Smoothed Projected Landweber for image reconstruction.

## 2. Background

### 2.1. Compressed Sensing Theory

The theory of compressed sensing projects a signal with sparse characteristics onto a low-dimensional space to realize the sampling and compression of the signal at the same time [13]. It accurately reconstructs the original signal with fewer sample values, and its basic model is as follows:(1)$$\begin{array}{c} y=\Phi x \end{array}$$
where $y\in R^{m}$ represents the result of compressed sampling, $\Phi \in R^{m\times n}$ represents the observation matrix, and $x\in R^{n}$ represents the original signal. The crux of the problem of CS is how to reconstruct the original signal *x* accurately from the compressed signal *y*.

The signal *x* in CS theory must be sparse [14]. In this way, the observed values after a random sub-sampling of the observation matrix can break the Nyquist sampling theorem and be reconstructed accurately. Most natural signals in the time domain are non-sparse but may be sparse in some transformation domains. That is to say that non-sparse signals can be converted into sparse signals using the sparse transformation basis [15]:(2)$$\begin{array}{c} x=\Psi s \end{array}$$
where $\Psi \in R^{n\times n}$ is the sparse transformation basis and $s\in R^{n}$ is the signal after sparse transformation. At this point, the basic model of CS can be expressed as follows:(3)$$\begin{array}{c} y=\Phi x=\Phi \Psi s=As \end{array}$$
where A=ΦΨ is the perception matrix that is the product of the observation matrix and the sparse basis; this model meets the requirements of CS theory. Sparse signals *s* can be obtained using the reconstruction algorithm, and then reconstructed signals *x* can be obtained using the sparse transformation x=Ψs, which is the reconstruction process of compressed sensing.

The common sparse transformation basis includes the discrete Fourier transform basis (DFT) [16], discrete cosine transform basis (DCT) [17], and discrete wavelet transform basis (DWT) [18]. The DCT can reflect the correlation of image signals, so it is mainly used for sparse representations of image signals.

In (Equation 3), this paper takes the random Gaussian measurement matrix as the observation matrix and the discrete cosine transform basis as the sparse basis to verify the performance of these reconstruction algorithms under the same conditions.

### 2.2. Compressed Imaging System and Reconstruction

Different from traditional image compression, which compresses and encodes complete image data, the compressed imaging system combines compressed sensing theory into the traditional optical imaging system and collects incomplete data. In this system, a coding mask is added to the imaging focal plane to encode the target scene, which is then collected by a low-resolution detector. The reconstruction algorithm is used to reconstruct the low-resolution images observed many times to obtain the corresponding high-resolution target scene. Its optical imaging system is shown in Figure 1. The imaging principle can be summarized as a light emphasis on the encoding mask and a down-sampling of the detector. Finally, the low-resolution image is the obtained observation value that adds detector noise. It can be reconstructed using the theory in Section 2.1.

The original signal *x* in (Equation 3) is a one−dimensional signal. This must be converted when applied to the compressed imaging system in Figure 1.

For a two−dimensional image signal of size M×M and a detector of size N×N, the single sampling process is as follows:Coding modulation. The coding board used in this system is a 0–1 coding board. The 0 position corresponds to no light, and the 1 position corresponds to light.down-sampling. The size of the image block of each pixel of the detector detected is B×B, where B=N/M, and the sum of the corresponding image block signals is the detection value of the detector.

It is expanded into a one−dimensional column vector $x\in R^{B^{2}\times 1}$ for the image block and into a one−dimensional row vector $\psi \in R^{1\times B^{2}}$ for the corresponding block of the encoding mask. We discover that y=ϕx represents a single sampling process of the corresponding image block position. Each image block uses the same encoding mask block, and all image encoding processes can be expressed as follows:(4)$$\begin{array}{c} y_{i}=\phi x_{i},i=1,2,\ldots ,M^{2} \end{array}$$
where $y_{i}\in R^{1}$ is the observation value of the i−th detector pixel.

The result of sampling the compression imaging system $pB^{2}$ times, using a different coding mask each time, is as follows:(5)$$\begin{array}{c} Y_{i}=\Phi X_{i},i=1,2,\ldots ,M^{2} \end{array}$$
where $Y_{i}\in R^{pB^{2}\times 1}$ are the observations of the i−th detector pixel, $X_{i}\in R^{B^{2}\times 1}$ is the column vector of the image block signal, and $\Phi \in R^{pB^{2}\times B^{2}}$ is the observation matrix. In this case, the compressed sensing reconstruction algorithm can be used for super-resolution from low-resolution images obtained by multiple sampling into high-resolution images. This is because the model in (Equation 5) meets the compressed sensing theoretical model in (Equation 3). This process is the multi-image super-resolution reconstruction technology based on compressed sensing theory.

### 2.3. BCS-SPL Algorithm

As can be seen in Section 2.2, the core of the compressed imaging reconstruction lies in block compressed sensing (BCS) [19]. The idea of the block also brings some adverse effects, such as the block effect in the reconstructed image. The block effect of the image under a DCT transformation is pronounced. Therefore, [12] proposed the Smooth Projection Landweber (SPL) algorithm, which deals with the image’s block effect and noise. The core idea of the SPL algorithm is two−step iterations starting from the initial sparseness on the transform domain. The process is as follows:(6)$$\begin{array}{c} \theta^{\left[n+1\right]}=\theta^{\left[n\right]}+\frac{1}{\zeta }\left(\Psi^{-1}\Phi^{T}\right)\left(y-\Phi \Psi \theta^{\left[n\right]}\right) \end{array}$$
(7)$${\hat{\theta }}^{\left[n+1\right]}=\left\{\begin{array}{cc} \theta^{\left[n+1\right]}, & \left|\theta^{\left[n+1\right]}\right|\geq \tau \\ 0, & otherwise \end{array}\right.$$
where $\theta^{\left[n\right]}$ is the reconstructed signal in the n−th iteration, ζ is the largest singular value of $\Phi \Phi^{T}$, and $\tau =\lambda \sqrt{2log_{e}N}\frac{MED\left(\left|{\hat{\theta }}^{\left[n\right]}\right|\right)}{0.6754}$ is the iteration threshold.

Mun S. et al. [12] proposed the reconstruction algorithm called BCS-SPL by applying the SPL algorithm to CS theory, which can remove block effects and impose smoothness. The BCS-SPL algorithm has been verified to have a good reconstruction effect, and its steps are as follows:Obtain the initial solution as $x_{i}^{\left[0\right]}=\Phi^{-1}y$;Smooth the reconstructed signal of the n−th iteration as $x_{i}^{\left[n\right]}=wiener\left(x_{i}^{\left[n\right]}\right)$, where wiener() is the Wiener filtering;Calculate according to (Equation 6) as ${\hat{x_{i}}}^{\left[n\right]}=x_{i}^{\left[n\right]}+\Phi^{T}\left(y_{i}-\Phi x_{i}^{\left[n\right]}\right)$;Transform the reconstructed signal to thw Ψ domain and obtain ${\hat{s_{i}}}^{\left[n\right]}=\Psi^{-1}{\hat{x_{i}}}^{\left[n\right]}$;Calculate according to (Equation 7) as
$${\hat{s_{i}}}^{\left[n\right]}=\left\{\begin{array}{cc} {\hat{s_{i}}}^{\left[n\right]}, & \left|{\hat{s_{i}}}^{\left[n\right]}\right|\geq \tau \\ 0, & otherwise \end{array}\right.;$$Transform ${\hat{s_{i}}}^{\left[n\right]}$ back to the spatial domain ${\hat{x_{i}}}^{\left[n\right]}=\Psi {\hat{s_{i}}}^{\left[n\right]}$;Calculate according to (Equation 6) as $x_{i}^{\left[n+1\right]}={\hat{x_{i}}}^{\left[n+1\right]}+\Phi^{T}\left(y_{i}-\Phi {\hat{x_{i}}}^{\left[n\right]}\right)$;If the termination condition $\left|D^{\left[n+1\right]}-D^{\left[n\right]}\right|<10^{-4}$ is satisfied, where
$$D^{\left[n\right]}={∥x_{i}^{\left[n+1\right]}-x_{i}^{\left[n\right]}∥}_{2}/N$$
then the iteration terminates. Otherwise, go to Step 2.

### 2.4. SL0 Algorithm

The solution of (Equation 3) is essentially to reconstruct the original signal with the least number of non-zeros based on the observation value and the perception matrix to obtain the most sparse solution. After that, the original signal in the natural domain can be obtained by sparse transformation, and the CS algorithm can be summarized as follows:(8)$$\begin{array}{c} s=\min{∥s∥}_{0},s.t.y=As \end{array}$$
(9)x=Ψs
where ${∥s∥}_{0}=\sum_{i=1}^{n}{\left|s_{i}\right|}^{0}$ is the $l_{0}$ norm of *s*, which represents the number of non-zero elements in the vector and the sparsity of the vector. The sparsest solution can be obtained by solving its minimum value. However, it can be seen that this is an NP-hard problem and difficult to solve when *n* is large.

Using a smooth continuous function to approximate the $l_{0}$ norm of the original sparse signal discontinued was proposed for the SL0 algorithm by Mohimani et al. [7]. The smoothing function adopted is the standard Gaussian function:(10)$$\begin{array}{c} f_{\sigma }\left(s_{i}\right)=1-e^{-\frac{s_{i}^{2}}{2\sigma^{2}}} \end{array}$$

It is easier for us to derive the following:(11)$$\lim_{\sigma \to 0}f_{\sigma }\left(s_{i}\right)=\left\{\begin{array}{cc} 0, & s_{i}=0\\ 1, & s_{i}\neq 0 \end{array}\right.$$

After bringing $F\left(s\right)=\sum_{i=1}^{n}f_{\sigma }\left(s_{i}\right)$ into (Equation 9), the approximation of the $l_{0}$ norm can be obtained as follows:(12)$$\begin{array}{c} {∥s∥}_{0}=\sum_{i=1}^{n}{\left|s_{i}\right|}^{0}\approx n-\lim_{\sigma \to 0}F_{\sigma }\left(s\right) \end{array}$$
where σ is the smoothness factor, whose value determines the smoothness degree of $F_{\sigma }\left(s\right)$. The larger σ is, the smoother $F_{\sigma }\left(s\right)$ is, and the lower the degree of approximation to the $l_{0}$ norm is. Otherwise, the $l_{0}$ norm of the signal *s* can be approximated by (Equation 10) when σ is approximately equal to zero.

The model in (Equation 8) can be transformed into an optimization problem to solve continuous signals. On this basis, the fastest descent method and gradient projection principle are used to gradually approach the optimal solution of the continuous function through several iterations.

The search direction of the optimal value is expressed as follows:(13)$$\begin{array}{c} d=-\nabla F_{\sigma }\left(s\right)=\left[s_{1}e^{-\frac{s_{1}^{2}}{2\sigma^{2}}},s_{2}e^{-\frac{s_{2}^{2}}{2\sigma^{2}}},\ldots ,s_{n}e^{-\frac{s_{n}^{2}}{2\sigma^{2}}}\right] \end{array}$$

The effect called “sawtooth” will appear in the search for the optimal value of the fastest descent method, which causes the global optimal value not to be obtained and the estimation accuracy of the $l_{0}$ norm to reduce. For these reasons, the algorithm needs to be improved.

## 3. Materials and Methods

### 3.1. Construct an Approximate Estimation Function of the $l_{0}$ Norm

The accuracy of the approximate $l_{0}$ norm is the key factor for improving the effectiveness of the SL0 algorithm. The closer the continuous smoothing function constructed is to the $l_{0}$ norm, the more accurate the result of the algorithm reconstruction will be. In this paper, the CGSL0 algorithm uses the following inverse trigonometric fraction function. It is proposed as the approximate estimation function of the $l_{0}$ norm:(14)$$\begin{array}{c} f_{\sigma }\left(s_{i}\right)=\frac{4}{\pi }\arctan\left(\frac{s_{i}^{2}}{s_{i}^{2}+\rho \sigma^{2}}\right) \end{array}$$
where ρ and σ are the parameters of controlling the steepness of the smoothing function and $s_{i}$ is the component of the sparse vector *s*. Therefore, the $l_{0}$ norm is approximately expressed as follows:(15)$$\begin{array}{c} {∥s∥}_{0}=\sum_{i=1}^{n}{\left|s_{i}\right|}^{0}\approx \lim_{\sigma \to 0}F_{\sigma }\left(s\right)=\lim_{\sigma \to 0}\sum_{i=1}^{n}f_{\sigma }\left(s_{i}\right) \end{array}$$

As shown in Figure 2, the Gaussian function, hyperbolic tangent function, and compound trigonometric function are, respectively, used to approximate the SL0 algorithm [7], NSL0 algorithm [20], and DNSL0 algorithm [21]. It can be observed that in the case of ρ=0.05 and σ=0.01, for the interval [−0.1 0.1], the inverse trigonometric fraction function proposed in this paper is steeper than other algorithms and has a better approximation to the $l_{0}$ norm; that is, the sparsity of the signals can be expressed more accurately. It can improve the accuracy of the optimization algorithm.

### 3.2. CGSL0 Algorithm

The SL0 algorithm has a “sawtooth” effect using the Gaussian function to approximate the $l_{0}$ norm. The NSL0 algorithm is solved by the modified Newton method. It needs to calculate the first- and second-order derivatives of the iterative points. At the same time, the Hesse matrix of the objective function must be positive definite, which has high requirements for the objective function. The inverse matrix of the second-order Hesse matrix requires a large amount of calculation, which presents some problems for practical use.

The conjugate gradient method is an unconstrained optimization method between the fastest descent method and the Newton method. It has a superlinear convergence speed, a simple algorithm structure, and easy programming implementation. This method only uses the first derivative to avoid the calculation of the second derivative, reducing the amount of calculation and storage, much like the fastest descent method. For the sake of the above, we propose an algorithm combining the smoothing function and the conjugate gradient method in Section 3.1, BCS-CGSL0.

According to (Equation 14), the reconstruction model of the CS algorithm is as follows:(16)$$\begin{array}{c} s=\min\sum_{i=1}^{n}\frac{4}{\pi }\arctan\left(\frac{s_{i}^{2}}{s_{i}^{2}+\rho \sigma^{2}}\right),s.t.y=As \end{array}$$

The solution can be divided into two steps:Calculate the iteration direction using the conjugate gradient method and search for the optimal value;Project the results of the conjugate gradient method into the feasible set using constraints.

#### 3.2.1. Conjugate Gradient Method to Find the Optimal Solution

The idea of the conjugate gradient method is to generate the conjugate direction of the Hesse matrix of the convex quadratic function by using the fastest descending direction at the current step at each iteration step. Firstly, choose the least squares solution as the initial value:(17)$$\begin{array}{c} s_{0}={\left(A^{T}A\right)}^{-1}A^{T}y \end{array}$$

The gradient direction of the fastest descent can be obtained according to (Equation 14):(18)$$\begin{array}{cc} g_{k}=\nabla F_{\sigma }\left(s\right) & =\left[\frac{\partial f_{\sigma }\left(s_{1}\right)}{\partial s_{1}},\ldots ,\frac{\partial f_{\sigma }\left(s_{1}\right)}{\partial s_{1}}\right]\\ \; & =\left[\frac{\rho \sigma^{2}s_{1}}{s_{1}^{2}+\rho \sigma^{2}s_{1}^{2}+\rho^{2}\sigma^{4}},\ldots ,\frac{\rho \sigma^{2}s_{1}}{s_{n}^{2}+\rho \sigma^{2}s_{n}^{2}+\rho^{2}\sigma^{4}}\right] \end{array}$$

In this method, we assume that the current iteration value is $x_{k}$ and the next iteration value is $x_{k+1}=x_{k}+\alpha_{k}d_{k}$, where $\alpha_{k}$ is the step of the current iteration and $d_{k}$ represents the search direction of the minimum value. The conjugate gradient method can then be expressed as follows:(19)$$d_{k}=\left\{\begin{array}{cc} -g_{k}, & k=1\\ -g_{k}+\beta_{k}d_{k-1}, & k>1 \end{array}\right.$$
where $g_{k}$ represents the gradient of the function and $\beta_{k}$ can be solved using different conjugate gradient methods. The classical solution expressions are as follows:(20)$$\beta_{k}^{FR}=\frac{{∥g_{k}∥}^{2}}{{∥g_{k-1}∥}^{2}},\beta_{k}^{PRP}=\frac{g_{k}^{T}\left(g_{k}-g_{k-1}\right)}{{∥g_{k-1}∥}^{2}},\beta_{k}^{DY}=\frac{{∥g_{k}∥}^{2}}{d_{k-1}^{T}\left(g_{k}-g_{k-1}\right)}$$

Different gradient conjugate methods usually have different performance results in different scenarios, among which FR and PRP conjugate gradient methods are more commonly used. Therefore, the hybrid conjugate gradient method can be used to modify it to a certain extent. In this paper, the conjugate gradient method of the FR and PRP hybrid is adopted, and the hybridization method is shown in (Equation 19):(21)$$\beta_{k}=\left\{\begin{array}{cc} \beta_{k}^{PRP}, & 0\leq \beta_{k}^{PRP}\leq \beta_{k}^{FP}\\ \beta_{k}^{FR}, & otherwise \end{array}\right.$$

The conjugate gradient method of the FR and PRP hybridization, which can avoid the disadvantage of producing continuous small steps, is an algorithm with a better all around performance among conjugate gradient methods. For this reason, it is selected as the search direction for calculating the minimum value.

#### 3.2.2. Project the Optimal Solution into the Feasible Set

The solution obtained above is *s*, which needs to be projected as follows to ensure that the solution is in the feasible set $\left\{X|Y=AX\right\}$ limited by the constraints.
(22)$$\begin{array}{cc} s_{p} & =\min{∥s_{p}-s∥}_{2},s.t.s_{p}\in X\\ \; & =s-A^{T}{\left(AA^{T}\right)}^{-1}\left(As-y\right) \end{array}$$
where $s_{p}$ is the solution in the feasible set.

### 3.3. BCS-CGSL0 Algorithm

Combining the BCS-SPL algorithm introduced in Section 2.3 and the CGSL0 algorithm introduced in Section 3.2, we propose a BCS-CGSL0 algorithm, which presents the iterative idea of SL0 under the framework of BCS-SPL. Not only does the BCS-CGSL0 algorithm ensure the accuracy and efficiency of reconstruction, but it also removes the block effect caused by block compressed sensing. The step-by-step process of the BCS-CGSL0 algorithm is as Algorithm 1:
**Algorithm 1:** BCS-CGSL0 Algorithm**Input**: measure signal *y*, measurement matrix Φ, transform domain basis Ψ, block size *B***initialization:**   $x_{i}^{\left[0\right]}=\Phi^{-1}y_{i}$;   $\sigma =[\sigma_{1},\sigma_{2},\ldots ,\sigma_{J}]$, where $\sigma_{j}=\rho \sigma_{j-1},0<\rho <1$ and $\sigma_{1}=2\max\left|{\hat{s_{i}}}^{\left[0\right]}\right|$;**for**j=0,1,…,J−1**do**   ${\hat{x}}_{i}^{\left[j\right]}=wiener\left(x_{i}^{\left[j\right]}\right)$, where wiener() represents the smoothing filter;   ${\tilde{x}}_{i}^{\left[j\right]}={\hat{x}}_{i}^{\left[j\right]}+\Phi^{T}\left(y-\Phi {\hat{x}}_{i}^{\left[j\right]}\right)$ and $s_{i}^{\left[j\right]}=\Psi^{-1}{\tilde{x}}_{i}^{\left[j\right]}$;   $\sigma =\sigma_{j}$ and $s=s_{i}^{\left[j\right]}$;    **for**
k=1,2,…,L
**do**
    $g_{k}=\left[\frac{\rho \sigma^{2}s_{1}}{s_{1}^{4}+\rho \sigma^{2}s_{1}^{2}+\rho^{2}\sigma^{4}},\cdots ,\frac{\rho \sigma^{2}s_{n}}{s_{n}^{4}+\rho \sigma^{2}s_{n}^{2}+\rho^{2}\sigma^{4}}\right]$, where $n=B^{2}$    $d_{k}=\left\{\begin{array}{cc} -g_{k}, & k=1\\ -g_{k}+\beta_{k}d_{k-1}, & k>1 \end{array}\right.$, where
$\beta_{k}=\left\{\begin{array}{cc} \beta_{k}^{PRP}, & 0\leq \beta_{k}^{PRP}\leq \beta_{k}^{FP}\\ \beta_{k}^{FP}, & otherwise \end{array}\right.$;    $s\leftarrow s+\mu d_{k}$ and
$s_{p}\leftarrow s-A^{T}{\left(AA^{T}\right)}^{-1}\left(As-y\right)$;   **end**   ${\hat{s}}_{i}^{\left[j\right]}=s$ and
${\hat{s}}_{i}^{\left[j\right]}=\left\{\begin{array}{cc} {\hat{s}}_{i}^{\left[j\right]}, & \left|{\hat{s}}_{i}^{\left[j\right]}\right|\geq \tau \\ 0, & otherwise \end{array}\right.$;   ${\overset{ˇ}{x}}_{i}^{\left[j\right]}=\Psi {\hat{s}}_{i}^{\left[j\right]}$ and $x_{i}^{\left[j\right]}={\overset{ˇ}{x}}_{i}^{\left[j\right]}+\Phi^{T}\left(y_{i}-\Phi {\overset{ˇ}{x}}_{i}^{\left[j\right]}\right)$;**end**(**Output**: the reconstructed image $x=x_{i}^{\left[j\right]}$

## 4. Experiments and Results

In the experiment, the size of the original images is 512×512. Due to the characteristics of block compressed sensing, a too large block size will greatly increase the reconstruction time, and a too small block size will reduce the reconstruction accuracy. Therefore, we choose the block size B=16 after comprehensive consideration. The observation matrix is an orthogonalized random Gaussian matrix. The sparsity transform basis is a discrete cosine transform (DCT). All experiments were performed using MATLAB 2022a on a computer equipped with an Intel Core TM i9, 3.7 GHz processor, with 32GB of RAM and running on Windows 10. In the BCS-CGSL0 algorithm, we set the decreasing factor as ρ=0.6, the iteration number as J=200,L=3, and the step size as μ=0.01. The threshold τ can be calculated according to (Equation 7), where λ=6.

Two sets of experiments are described in this paper. The first experiment compares the reconstruction effects of the BCS-CGSL0 and SL0 series algorithms at a 0.1–0.5 sampling rate. The second experiment compares the reconstruction effects of the BCS-CGSL0 and no-SL0 series algorithms at a 0.1–0.5 sampling rate. Figure 3 presents the original and observed images of Goldhill and Clown.

The evaluation indicator in the experiment includes the peak signal-to-noise ratio (PSNR), structural similarity (SSIM), and reconstruction time. The results of the two groups of experiments are in Figure 3.

### 4.1. The Comparison of the BCS-CGSL0 and SL0 Series Algorithms

In the first experiment, we evaluated the effectiveness of the proposed BCS-CGSL0 and SL0 series of algorithms, including the SL0 algorithm [7], NSL0 algorithm [20], DNSL0 algorithm [21], and CGSL0 algorithm proposed in this paper.

Table 1 lists these algorithms’ PSNR, SSIM, and reconstruction time for the Goldhill and Clown images when the rate is 0.5. As can be seen, the SL0 series algorithms have a fast reconstruction speed while ensuring a high reconstruction accuracy. Compared with the other four algorithms, the BCS-CGSL0 algorithm proposed in this paper reduces the reconstruction time while improving the reconstruction accuracy (PSNR and SSIM). The BCS-CGSL0 algorithm has the best effect and reasonable practicability when combining the three evaluation indicators.

Table 2 lists the reconstruction effect of these algorithms for the Goldhill image when the rate is between 0.1 and 0.5. It can be seen that the effect of the SL0 series algorithm is very poor at low sampling rates. For example, the reconstruction effect of the SL0 algorithm is worse when the rate is 0.1, making the reconstructed image unusable. This demonstrates that the reconstruction accuracy of the BCS-CGSL0 algorithm is the highest; the PSNR is increased by more than 2 dB, and the SSIM is higher than the other four algorithms.

Figure 4 compares the PSNR and SSIM of the BCS-CGSL0 and SL0 series algorithms for the Goldhill image when the rate is between 0.1 and 0.8. It can be seen that the SL0 series algorithms are greatly affected by the sampling rate, and the effect is extremely poor when the sampling rate is low. However, the BCS-CGSL0 algorithm overcomes this shortcoming and achieves good reconstruction results when the rate is 0.1, making the BCS-CGSL0 algorithm adaptable to more scenarios.

### 4.2. The Comparison of The BCS-CGSL0 and non-SL0 Series Algorithms

In the second experiment, we evaluated the effectiveness of the proposed BCS-CGSL0 and no-SL0 series of algorithms, including OMP algorithm [6], Split-Bregman algorithm [9], IRLS algorithm [10], FOCUSS algorithm [11], BCS-SPL algorithm [12], and BCS-TVAL3 algorithm [22]. Table 3 lists these algorithms’ PSNR, SSIM, and reconstruction time for the Goldhill and Clown images when the rate is 0.5. Compared with Table 1, it can be seen that these typical no-SL0 series algorithms have a good reconstruction accuracy. However, most take a long time and are unsuitable for real-time scenarios. In contrast, the BCS-CGSL0 algorithm has good practicability since it can complete the reconstruction in a short time and have a better reconstruction accuracy.

Table 4 lists the reconstruction effect of these algorithms for the Goldhill image when the rate is between 0.1 and 0.5. Compared with Table 1, it can be seen that the no-SL0 series algorithms still have a relatively stable reconstruction effect at low resolutions, which is better than the SL0 series algorithms. However, their disadvantage is that the reconstruction time is too long for adaptation to real-time scenarios. Although the BCS-SPL algorithm’s reconstruction speed is fast, the reconstruction accuracy still needs to be improved. In contrast, the advantages of the BCS-CGSL0 algorithm are more obvious, including a higher reconstruction accuracy and shorter reconstruction time.

Figure 5 compares the PSNR and SSIM of the BCS-CGSL0 and no-SL0 series algorithms for the Goldhill image when the rate is between 0.1 and 0.8. It intuitively demonstrates that the reconstruction effect of no-SL0 series algorithms is more stable at different sampling rates and that the BCS-CGSL0 algorithm has an improved performance.

## 5. Conclusions

This study proposes a compressed sensing reconstruction algorithm based on block compressed sensing with a conjugate gradient smoothed $l_{0}$ norm, BCS-CGSL0. The core of our method is to combine the idea of the SL0 series algorithms with the BCS-SPL algorithms, to remove the block effect of the reconstructed image and improve the reconstruction accuracy and speed. The main contributions of this paper are as follows:We propose a new function called the inverse trigonometric fraction function, which approximates the $l_{0}$ norm better than similar functions;We propose a method for optimizing the SL0 algorithm (CGSL0), using the inverse trigonometric fraction function to approximate the $l_{0}$ norm and the modified conjugate gradient method to solve the optimization problem;We propose a reconstruction algorithm that combines CGSL0 and BCS-SPL, which has a high reconstruction accuracy and removes the blockiness of reconstructed images.

Through the simulation experiment verification of encoding low-resolution images, compared with SL0 series and no-SL0 series algorithms, this algorithm can ensure a good reconstruction speed when improving the reconstruction accuracy, which ensures that the algorithm has great value in practice. In future work, we plan to further study improvements to the SL0 series of algorithms, improve their accuracy and speed, and enhance their applicability in practical scenarios.

## Figures and Tables

**Figure 1 sensors-23-04870-f001:**
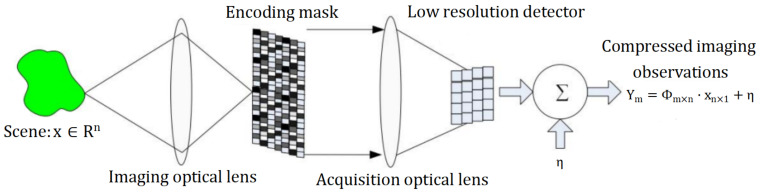
Compressed imaging system.

**Figure 2 sensors-23-04870-f002:**
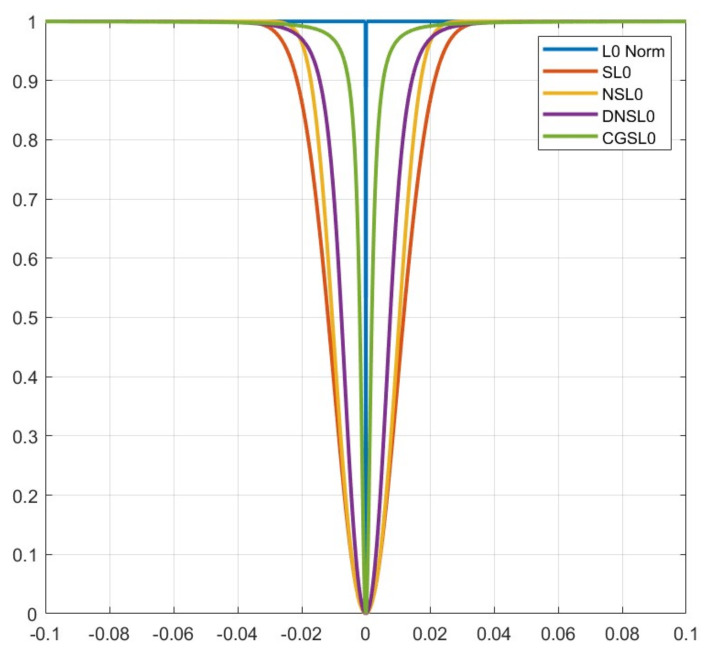
Smooth function model comparison diagram. The most central curve is the $l_{0}$ norm. The remaining curves from inside to outside are smooth function approximation models of SL0, NSL0, DNSL0, and CGSL0 algorithms. It can be seen that the CGSL0 smooth function has the highest approximation to the $l_{0}$ norm.

**Figure 3 sensors-23-04870-f003:**
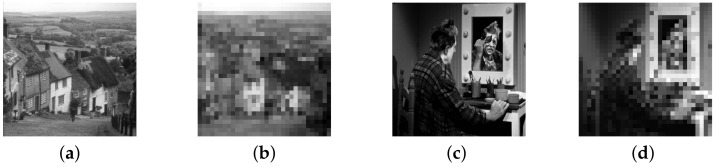
Images used in the experiment and their observations: (**a**) original image of Goldhill with a size of 512 × 512; (**b**) observed image of Goldhill with a size of 32 × 32; (**c**) original image of Clown with a size of 512 × 512; (**d**) observed image of Clown with a size of 32 × 32.

**Figure 4 sensors-23-04870-f004:**
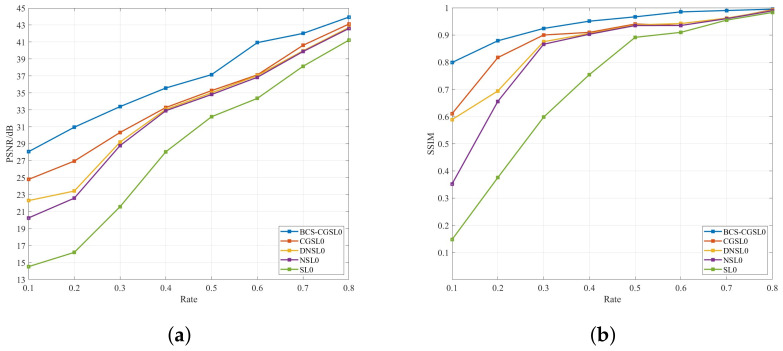
Comparison of the reconstruction effect of the BCS-CGSL0 and SL0 series algorithms for Goldhill image: (**a**) comparison of PSNR; (**b**) comparison of SSIM.

**Figure 5 sensors-23-04870-f005:**
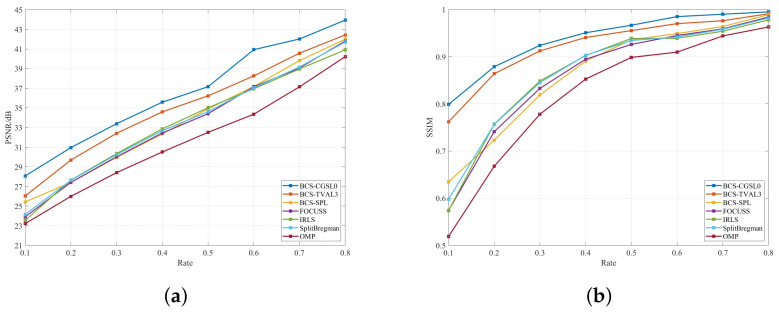
Comparison of the reconstruction effect of the BCS-CGSL0 and no-SL0 series algorithms for Goldhill image: (**a**) comparison of PSNR. (**b**) comparison of SSIM.

**Table 1 sensors-23-04870-t001:** Comparison of reconstruction quality for Goldhill and Clown images with SL0, NSL0, DNSL0, CGSL0, and BCS-CGSL0 algorithms.

Image	Algorithm	PSNR/dB	SSIM	Tims/s
Goldhill	SL0	32.1960	0.8913	0.5027
	NSL0	34.8157	0.9347	0.9250
	DNSL0	35.0088	0.9371	2.1053
	CGSL0	35.2736	0.9403	0.8343
	BCS-CGSL0	37.5592	0.9667	0.8078
Clown	SL0	30.9515	0.9038	0.5037
	NSL0	34.6306	0.9542	0.9064
	DNSL0	34.9998	0.9572	1.9203
	CGSL0	35.6138	0.9617	0.7518
	BCS-CGSL0	38.5311	0.9778	0.8574

**Table 2 sensors-23-04870-t002:** Reconstruction effect of SL0, NSL0, DNSL0, CGSL0, and BCS-CGSL0 algorithms for Goldhill image when the rate is between 0.1 and 0.5.

Sample Rate	Algorithm	PSNR/dB	SSIM	Tims/s
0.1	SL0	14.5255	0.1474	0.2132
	NSL0	20.2683	0.3517	0.2832
	DNSL0	22.3120	0.5887	1.1115
	CGSL0	24.8058	0.6104	1.1550
	BCS-CGSL0	28.0709	0.7988	1.0696
0.2	SL0	16.2064	0.3754	0.2998
	NSL0	22.5954	0.6557	0.3816
	DNSL0	23.4365	0.6940	1.3254
	CGSL0	26.9654	0.8169	1.2732
	BCS-CGSL0	30.9586	0.8787	0.9225
0.3	SL0	21.5878	0.5976	0.4207
	NSL0	28.7902	0.8659	0.6539
	DNSL0	29.2017	0.8752	1.6780
	CGSL0	30.3402	0.8999	1.6083
	BCS-CGSL0	33.3883	0.9240	1.1859
0.4	SL0	28.0617	0.7542	0.4612
	NSL0	32.9041	0.9027	0.7291
	DNSL0	33.0616	0.9056	1.8192
	CGSL0	33.2796	0.9096	0.7126
	BCS-CGSL0	35.5813	0.9508	0.9762
0.5	SL0	32.1960	0.8913	0.5027
	NSL0	34.8157	0.9347	0.9250
	DNSL0	35.0088	0.9371	2.1053
	CGSL0	35.2736	0.9403	0.8343
	BCS-CGSL0	37.5592	0.9667	0.8078

**Table 3 sensors-23-04870-t003:** Comparison of the reconstruction quality for Goldhill and Clown images with the OMP, IRLS, Split-Bregman, BCS-SPL, BCS-TVAL3, and BCS-CGSL0 algorithms.

Image	Algorithm	PSNR/dB	SSIM	Tims/s
Goldhill	OMP	32.5068	0.8985	5.2730
	Split-Bregman	34.6123	0.9343	13.0204
	IRLS	35.0187	0.9387	32.6246
	FOCUSS	34.4092	0.9260	92.4273
	BCS-SPL	34.9058	0.9356	0.4938
	BCS-TVAL3	35.6239	0.9320	8.4257
	BCS-CGSL0	37.5592	0.9667	0.8078
Clown	OMP	33.3586	0.9405	5.3338
	Split-Bregman	34.6607	0.9561	11.8532
	IRLS	35.2617	0.9616	32.4094
	FOCUSS	34.4967	0.9533	79.2619
	BCS-SPL	35.3601	0.9348	0.7055
	BCS-TVAL3	36.2245	0.9553	9.9349
	BCS-CGSL0	38.5311	0.9778	0.8574

**Table 4 sensors-23-04870-t004:** The reconstruction effect of the OMP, Split-Bregman, IRLS, FOCUSS, BCS-SPL, BCS-TVAL3, and BCS-CGSL0 algorithms for Goldhill image when the rate is between 0.1 and 0.5.

Sample Rate	Algorithm	PSNR/dB	SSIM	Tims/s
0.1	OMP	23.2151	0.5187	0.2837
	Split-Bregman	24.1545	0.5978	110.5349
	IRLS	23.5454	0.5734	8.0813
	FOCUSS	23.9018	0.5737	20.9178
	BCS-SPL	25.4181	0.6345	0.8154
	BCS-TVAL3	26.0347	0.7618	10.8333
	BCS-CGSL0	28.0709	0.7988	1.0696
0.2	OMP	25.9941	0.6677	0.7621
	Split-Bregman	27.6278	0.7564	55.2424
	IRLS	27.6465	0.7571	12.6816
	FOCUSS	27.4172	0.7411	40.7951
	BCS-SPL	27.4142	0.7231	0.7601
	BCS-TVAL3	29.6901	0.8639	10.0676
	BCS-CGSL0	30.9586	0.8787	0.9225
0.3	OMP	28.4080	0.7781	1.7354
	Split-Bregman	30.2447	0.8451	32.7416
	IRLS	30.3540	0.8485	18.1794
	FOCUSS	30.0229	0.8329	56.2880
	BCS-SPL	29.9553	0.8191	0.7631
	BCS-TVAL3	32.4181	0.9123	9.8397
	BCS-CGSL0	33.3883	0.9240	1.1859
0.4	OMP	30.5107	0.8525	3.0058
	Split-Bregman	32.6824	0.9031	19.7576
	IRLS	32.8625	0.9023	24.7105
	FOCUSS	32.4433	0.8947	75.5162
	BCS-SPL	32.3505	0.8908	0.6507
	BCS-TVAL3	34.5926	0.9407	9.7414
	BCS-CGSL0	35.5813	0.9508	0.9762
0.5	OMP	32.5068	0.8985	5.2730
	Split-Bregman	34.6123	0.9343	13.0204
	IRLS	35.0187	0.9387	32.6246
	FOCUSS	34.4092	0.9260	92.4273
	BCS-SPL	34.9058	0.9356	0.4938
	BCS-TVAL3	36.2245	0.9553	9.9349
	BCS-CGSL0	37.5592	0.9667	0.8078

## Data Availability

Not applicable.

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
