# Peer review of "Compressed Imaging Reconstruction Based on Block Compressed Sensing with Conjugate Gradient Smoothed l0 Norm"

_sensors, 2023, doi:10.3390/s23104870_

Round 1

Reviewer 1 Report

The paper proposes a compressed coding  using a reconstruction algorithm based on block compressed sensing with a conjugate gradient smoothed l0 norm termed as BCS-CGSL0.

According to CS theory, the sparse representation, measurement matrix and reconstruction algorithm are the three key elements in reconstructing the original signal from the sparse signal with high probability. 

I think that the authors have done a satisfactory effort in doing the research. They have shown their method (BCS-CGSL0) outperforms the others methods (DNSL0 and CGSL0). But what about a variable block size like the benchmark compression standards JPEG2000, H264 or HEVC?

Can the authors elaborate the need and motivation for the proposed work? The block-based hybrid video coding standards such as JPEG, JPEG2000, H264,HEVC and VVC are dominating the current image/video compression domain, and it is important for the reader to explain the need for the proposed work over the current methods that in use.

It is better if the authors can highlight the applications of compression sensing based block coding. For example, broadcasting industry uses block based hybrid video coding such as H264, HEVC etc. Is compression sensing based block coding suitable for such applications?

It is important that authors compare the results (Table1 and Table3, Figure 4 and Figure 5) of the proposed algorithm BCS-CGSL0 with standards such as JPEG or JPEG2000.

Some suggestions are as follows:

  1. In order to make the method more clearly and more attractive for the compression community, it is better to provide a link (or some comparaison) between this method (BCS-CGSL0) and the actuel standards (JPEG, JPEG2000).
  2. Please be sure to give all the parameters used in your system. It would even be preferred if you could add the parameters to a weblink (GetHub) to your system (either source code or executable!)
  3. The authors make certain claims during the discussion of the experimental results that either needs more clarification/explanation or needs some referencing;

Reviewer 2 Report

This paper proposes a compressed sensing reconstruction algorithm based on block compressed sensing, which has the conjugate gradient smoothing l0 norm BCS-CGSL0. The algorithm can be divided into two parts. In the first part, CGSL0 optimizes the SL0 algorithm, constructs a new inverse trigonometric fractional function to approximate the l0 norm, and uses the improved conjugate gradient method to solve the optimization problem. In the second part, the BCS-SPL method combined with the block compressed sensing framework removes the block effect. In short, the core of the method used in this paper is to combine the ideas of SL0 series algorithms with BCS-SPL algorithms to remove the block effect of reconstructed images and improve the accuracy and speed of reconstructed images.

However, there are some problems with this article. For example, in terms of formatting, there were some problems with the numbering of formulas that needed to be modified. It is recommended that the four pictures in Figure 3 also be marked with serial numbers, and the format of the references should be carefully checked and revised.

Reviewer 3 Report

The manuscript topic is very interesting. Here are some comments:

1. Fig2 &3 captions need to be added.

2. pseudocode can be improved

3. fig4 and table2 provide the same information as well as fig5 and table4

4. The block size 16. Did you try different values beside this value?

Round 2

Reviewer 1 Report

The authors should compare theirs results with:

  1. Ebrahim, M.; Chong, C.W.; Adil, S.H.; Raza, K. Block Compressive Sensing (BCS) Based Low Complexity, Energy Efficient Visual Sensor Platform with Joint Multi-Phase Decoder (JMD). Sensors 201919, 2039. [Google Scholar] [CrossRef] [PubMed][Green Version]
  2. Ebrahim, M.; Adil, S.H.; Raza, K.; Ali, S.S.A. Block Compressive Sensing Single-View Video Reconstruction Using Joint Decoding Framework for Low Power Real Time Applications. Appl. Sci. 202010, 7963. https://doi.org/10.3390/app10227963
  3.  
